# The Plant Growth-Promoting Potential of Halotolerant Bacteria Is Not Phylogenetically Determined: Evidence from Two *Bacillus megaterium* Strains Isolated from Saline Soils Used to Grow Wheat

**DOI:** 10.3390/microorganisms11071687

**Published:** 2023-06-28

**Authors:** Sylia Ait Bessai, Joana Cruz, Pablo Carril, Juliana Melo, Margarida M. Santana, Abdul M. Mouazen, Cristina Cruz, Ajar Nath Yadav, Teresa Dias, El-hafid Nabti

**Affiliations:** 1Laboratoire de Maitrise des Energies Renouvelables, Faculté des Sciences de la Nature et de la Vie, Université de Bejaia, Bejaia 06000, Algeria; aitsylia@gmail.com (S.A.B.); nabtielhafid1977@yahoo.com (E.-h.N.); 2cE3c–Centre for Ecology, Evolution and Environmental Changes and CHANGE–Global Change and Sustainability Institute, Faculdade de Ciências, Universidade de Lisboa, Campo Grande, 1749-016 Lisboa, Portugal; joana.ccruz@gmail.com (J.C.); paoloypunto_3@hotmail.com (P.C.); jmdconceicao@ciencias.ulisboa.pt (J.M.); mmcsantana@ciencias.ulisboa.pt (M.M.S.); ccruz@fc.ul.pt (C.C.); 3Competence Centre for Molecular Biology, SGS Molecular, Polo Tecnológico de Lisboa, Rua Cesina Adães Bermudes, Lt 11, 1600-604 Lisboa, Portugal; 4Department of Environment, Faculty of Bioscience Engineering, Ghent University, 9000 Gent, Belgium; abdul.mouazen@ugent.be; 5Department of Biotechnology, Dr. Khem Singh Gill Akal College of Agriculture, Eternal University, Baru Sahib, Sirmour 173101, India; ajar@eternaluniversity.edu.in

**Keywords:** biofertilizer, halotolerant bacterial strains, plant growth promoting traits, salinity, wheat

## Abstract

(1) Background: Increasing salinity, further potentiated by climate change and soil degradation, will jeopardize food security even more. Therefore, there is an urgent need for sustainable agricultural practices capable of maintaining high crop yields despite adverse conditions. Here, we tested if wheat, a salt-sensitive crop, could be a good reservoir for halotolerant bacteria with plant growth-promoting (PGP) capabilities. (2) Methods: We used two agricultural soils from Algeria, which differ in salinity but are both used to grow wheat. Soil halotolerant bacterial strains were isolated and screened for 12 PGP traits related to phytohormone production, improved nitrogen and phosphorus availability, nutrient cycling, and plant defence. The four ‘most promising’ halotolerant PGPB strains were tested hydroponically on wheat by measuring their effect on germination, survival, and biomass along a salinity gradient. (3) Results: Two halotolerant bacterial strains with PGP traits were isolated from the non-saline soil and were identified as *Bacillus subtilis* and *Pseudomonas fluorescens*, and another two halotolerant bacterial strains with PGP traits were isolated from the saline soil and identified as *B. megaterium*. When grown under 250 mM of NaCl, only the inoculated wheat seedlings survived. The halotolerant bacterial strain that displayed all 12 PGP traits and promoted seed germination and plant growth the most was one of the *B. megaterium* strains isolated from the saline soil. Although they both belonged to the *B. megaterium* clade and displayed a remarkable halotolerance, the two bacterial strains isolated from the saline soil differed in two PGP traits and had different effects on plant performance, which clearly shows that PGP potential is not phylogenetically determined. (4) Conclusions: Our data highlight that salt-sensitive plants and non-saline soils can be reservoirs for halotolerant microbes with the potential to become effective and sustainable strategies to improve plant tolerance to salinity. However, these strains need to be tested under field conditions and with more crops before being considered biofertilizer candidates.

## 1. Introduction

Worldwide, salinity affects more than 20% of the arable land, and by 2050, the affected area is estimated to reach 50% [1,2,3]. Saline soils (i.e., soils which have an electrical conductivity for the saturation soil extract of more than 4 dS m^−1^ or ~40 mM of NaCl at 25 °C [4]) have negative effects on most plant species that are salt-sensitive. As salinity inhibits many physiological mechanisms (e.g., water and nutrient absorption, DNA replication, photosynthesis, respiration, protein metabolism) in salt-sensitive plant species (and even in some salt-tolerant or halophyte species) [5,6,7], the negative effects of high salinity manifest along the plant’s life cycle, from germination to the final growth stages, with plant growth and survival being affected [3,8,9,10]. For example, salinity can decrease the yield of important crops (e.g., wheat, maize, rice, and barley) by up to 70% [5].

Salinity can be further intensified by ongoing climate change [11,12,13]. Drylands (which include dry sub-humid, semiarid, arid, and hyper-arid areas) are among the most susceptible biomes to land degradation and climate change [14] due to their characteristic low and variable rainfall and poor soils [15]. Since drylands host 38% of the global human population [16], whose livelihoods are often tied to subsistence agriculture and livestock production [17], increased salinity further potentiated by climate change will further jeopardize food security for local populations. Altogether, the area of marginal lands (i.e., arable lands that became less productive due to increased salinity) is increasing, and so is the urgency for sustainable agricultural practices capable of maintaining high crop yields despite adverse conditions [18,19].

One important approach is to make use of plant growth-promoting bacteria (PGPB). PGPB can improve crop production with multiple modes of action such as the synthesis of growth-promoting substances (including phytohormones such as auxins, strigolactones, and nitric oxide), improved plant nutrition, and resistance to biotic and abiotic stresses [18,20,21,22,23], including the stimulation of plant defence [24,25]. Indeed, the benefits of PGPB for a plant may occur along the plant’s life cycle as they may improve plant germination, survival, health, growth, reproduction, and productivity under optimal and adverse conditions. Usually, PGPB screening and isolation are target-oriented toward the following traits: phytohormone production (e.g., auxins, strigolactones), atmospheric nitrogen fixation, phosphate solubilization, and secretion of enzymes involved in nutrient cycling to improve nutrient availability [26,27,28]. Recently, numerous studies reported that some PGPB (belonging to several genera such as *Bacillus*, *Pseudomonas*, *Rhizobium*, and *Streptomyces*) have the ability to increase plant tolerance to salinity for several crops such as maize, mung beans, potato, tomato, and wheat [29].

Phytohormones (or plant hormones or plant growth regulators) are organic compounds that influence plant physiology (e.g., cell division, root and stem elongation/inhibition, development of buds and branches, chlorophyll production) and plant microbiome assembly [30]. In addition to endogenous phytohormones (i.e., those produced by the plant itself), soil microbes constitute a source of exogenous phytohormones (e.g., *Rhizobium*, *Enterobacter*, *Bacillus*, and *Pseudomonas* genera) [29,31] as they produce and excrete these compounds, making them available and beneficial for plants [21]. In particular, auxin indoleacetic acid (IAA) has been shown to be involved in seed germination, tissue differentiation, leaf expansion, lateral and adventitious root initiation, root hair development (with positive consequences for plant water and nutrients uptake), and root and stem elongation as well as increasing plant resistance to stress conditions [31,32,33,34,35]. As salinity reduces plant phytohormone production, endogenous phytohormone levels decrease, which hampers seed germination and plant development and productivity [31,36]. Therefore, fostering the interaction between a plant host and PGPB strains producing phytohormones can compensate for the salinity-driven reduction in endogenous phytohormones and restore the positive effects of phytohormones on plant development and physiology (including seed germination and root proliferation) [29,31].

Furthermore, PGPB provide other important services to plants, such as enhanced nutrient uptake and protection against pests and diseases. Plant growth in terrestrial ecosystems is usually limited by the availability of nitrogen (N) and/or phosphorous (P) [37]. Plants contribute directly to their own nutrition by taking up nutrients through their roots (N, P, and the other nutrients) and indirectly by interacting with guilds (i.e., groups of species that have similar requirements and play similar roles within a community) of functional groups of microbes (including PGPB) living in their roots and the surrounding soil [15,21]. Therefore, PGPB presenting traits capable of improving plant N (e.g., atmospheric nitrogen fixation, ammonia production, and nutrient cycling, in general) and P nutrition (e.g., phosphate solubilization) will contribute to a positive effect on plant growth and development [5,21].

High salinity also increases a plant’s susceptibility to several phytopathogens and promotes some fungal soil-borne plant diseases [38], which further threatens plant growth and survival. However, PGPB can modulate plant host immunity with several ingenious mechanisms by which pathogenic and beneficial microbes in the plant microbiome communicate with their host [39]. For example, the presence of PGPB capable of producing hydrogen cyanide (HCN) and ammonia (NH_3_) can play a crucial role in the biocontrol of fungal phytopathogens through inhibition of mycelial growth. While the synthesis of HCN inhibits cytochrome C oxidase and other important metalloproteins [40], the release of NH_3_ by PGPB impairs the growth of certain fungi and inhibits the germination of several fungal spores [41]. Furthermore, some PGPB can control phytopathogens with other non-exclusive mechanisms including:

(i) The production of fungal cell wall-degrading enzymes such as lipase (can degrade some fungal cell wall-associated lipids), chitinase (can degrade the integral fungal cell wall component chitin), and protease (can degrade cell wall proteins). The activity of these extracellular enzymes also releases nutrients, which can contribute to improve plant nutrition [5];

(ii) The increased difficulty for the phytopathogens to proliferate due to biotic interactions within the rhizosphere (i.e., competition for nutrients or root binding sites, predation, and parasitism) [22].

Similar to plants, soil microbes differ in their halotolerance, i.e., tolerance to ionic stress, or the ability of an organism to grow in salt concentrations higher than those required for growth. In the case of bacteria, while non-halotolerant bacteria can only grow in low salt concentrations (up to 1% *w*/*v*), halotolerant bacteria can grow in the absence of salt and in the presence of high salt concentrations. Halotolerant bacteria can be: (i) slightly tolerant if they survive in up to 2–8% salt; (ii) moderately tolerant if they survive in up to 18–20% salt; and (iii) extremely tolerant if they can grow over the whole range of salt concentrations from zero to saturation [42]. Therefore, salinity acts as a strong environmental filter, selecting soil microbes based on their halotolerance, i.e., salinity reduces soil microbial biomass, diversity, and functioning, especially of non-halotolerant microbes [43]. In accordance, halotolerant bacteria with PGP traits isolated under the influence of halophyte plants have been shown to stimulate plant growth and increase the salinity tolerance of salt-sensitive crops [5]. However, the facts that PGPB isolated from a given crop may not be as efficient/beneficial to other crops [18,44,45] and that most important crops are salt-sensitive (e.g., maize, mung beans, potato, tomato, and wheat [29]) can hamper the development of biofertilizers that sustain high crop yields despite increasing salinity.

Therefore, our objective was to isolate halotolerant bacterial strains with PGP capabilities from a salt-sensitive crop and to assess their plant growth-promoting potential. We hypothesised that along the salinisation process that constitutes a strong environmental filter, and despite the negative effects of salinity on plant growth and development, even salt-sensitive crops will have recruited and/or promoted the growth of halotolerant bacteria with PGP traits. As a result of the continued and increasing salinity, we expected to find halotolerant bacterial strains with PGP capabilities in saline soils where salt-sensitive crops are cultivated. The salt-sensitive crop with global importance and impact we used was wheat (*Triticum aestivum* L.), which is a staple crop for 35% of the world population [46]; more than two-thirds of global wheat is used for food and one-fifth is used for livestock feed [47]. Furthermore, the global demand for wheat follows human population growth, but the per cent of arable land affected by salinity compromises wheat production in many countries. On the other hand, we hypothesised that halotolerant bacteria with PGP traits would be more abundant in saline soils and, therefore, we sampled two agricultural soils in Algeria, which are used to grow wheat but differ in salinity level. Algeria includes arid and semi-arid regions and is among the Mediterranean countries where long-term drought has led to soil salinization [48]. The two soils were characterised physico-chemically and used to isolate and screen halotolerant bacterial strains for 12 PGP traits related to phytohormone production, improved N and P availability, nutrient cycling, and plant defence. Finally, the bacterial strain candidates for the ‘best’ halotolerant PGPB were tested in vivo by inoculating them on wheat seeds and testing their effect on germination, survival, and biomass along a salinity gradient ranging from no salinity to extreme salinity [49].

## 2. Materials and Methods

### 2.1. Soil Sampling and Soil Physical and Chemical Characterisation

Soil samples were collected from two agricultural fields differing in salinity level (Table 1), where wheat (*Triticum aestivum* L.) is usually grown. The fields were located 200 to 500 m from the Mediterranean Sea in northern Algeria (Béjaïa and Algiers). Sampling occurred in March 2019, which corresponded to an interval between crops when wheat had already been harvested. Soil samples were collected at 0–20 cm depth, as described in Melo et al. [26]. Each agricultural field was sampled at three random points spaced at a distance > 2 m. Four soil subsamples were collected from around each sampling point and mixed to form one composite sample per sampling site. In total, we collected 2 agricultural fields × 3 points = 6 samples. Soil samples were collected in labelled plastic bags and placed in polystyrene boxes that kept the soil at 4 °C. Samples were transported to the laboratory, where they were immediately analysed.

Each soil sample was characterised for electrical conductivity (EC; soils with EC > 4 ds m^−1^ are considered saline), sodium (Na) concentration, pH, organic matter (OM), organic carbon (Org C), and concentrations of micronutrients (iron, copper, zinc, and manganese). Soil pH and EC were measured in a 1:10 (*w*/*v*) water extract using a selective electrode for H^+^ (Crison micro pH 2002) and a conductivity meter (Consort C562), respectively. Soil organic matter was determined using loss on ignition overnight at 600 °C according to ISO norm 10694. The carbon in the soil microbial biomass (Org C) was determined using the fumigation-extraction method [50]. Sodium and micronutrients were determined using inductively coupled plasma-optical emission spectroscopy (ICP-OES–Spectro Ciros CCD, Spectro, Kleve, Germany). These analyses were conducted at the Physico-chemical soil analysis laboratory, PROFER, Mostaganem, Algeria.

### 2.2. Screening for Halotolerant PGP Bacteria Candidates

Upon arrival at the laboratory, one gram of each soil sample was suspended and serially diluted (10^−1^ to 10^−8^) in PSB (phosphate saline broth). Next, 100 μL of each dilution was used to inoculate a sterile nutrient agar (NA) medium Petri dish (9 cm diameter), which was then incubated for 48 h at 30 ± 2 °C. Several bacterial colonies developed. When the colonies could be morphologically distinguished and were not overlapping with other colonies, they were re-inoculated (i.e., they were picked up and streaked) on sterile plates with NA to obtain pure cultures. The morphologically distinct bacterial isolates were kept at 4 °C for their physiological and genetic characterisation.

#### 2.2.1. Halotolerance

The tolerance to salinity of the bacterial isolates (i.e., halotolerance) was tested with increasing NaCl concentrations (0; 100; 200; 300; 400; 500; 600; 700; 800; 900; 1000; 1100; 1200; 1300; 1400; and 1500 mM) using glucose minimal medium (GMM) with the following composition in 1 L of distilled water: 5 g glucose; 1 g NH_4_Cl; 3 g KH_2_PO_4_; 2.4 g Na_2_HPO_4_; and 0.2 g MgSO_4_.7H_2_O at pH: 7.0 ± 0.2. For each NaCl concentration, the bacterial strains were inoculated with an initial OD_600_ = 0.1 and incubated under shaking (Orbital model 200) at 30 °C and 120 rpm for 48 h. Then, bacterial growth was measured as cell density determined using spectrophotometric reading at 600 nm (SHIMADZU UV-1800). Each NaCl concentration was tested in triplicate. The bacterial isolates that grew at NaCl concentration >100 mM (i.e., were halotolerant) were further tested.

#### 2.2.2. Production and Quantification of Indole Acetic Acid (IAA)

The halotolerant isolates (i.e., the bacterial isolates that grew at NaCl concentration > 100 mM) were assessed for their potential to synthesise IAA, using the Gordon and Weber [51] method. The bacterial strains were grown for 24 h until reaching OD_600_ = 0.5, after which they were inoculated in Luria Bertani (LB) liquid medium supplemented with 0.5 mg/mL of L-Tryptophan and 0.5% of glucose (both solutions were filter-sterilised). After inoculation in the LB medium, the bacterial strains were incubated on a rotary shaker at 120 rpm for 4 days at 30 ± 2 °C (Orbital model 200, Yohmai, Stains, France). After centrifugation for 10 min at 10,000× *g* (Biocen 22 R, Ortoalresa, Madrid, Spain), culture supernatants were collected and mixed with an equal volume of Salkowski’s reagent (2% of FeCl_3_–0.5 M–in 35% HClO_4_). The mixtures were incubated for 20 min in the dark at 25 ± 2 °C. The bacterial cultures capable of producing IAA turned pink, and the IAA quantification was performed by measuring the absorbance at 530 nm (SHIMADZU UV-1800, Kyoto, Japan). IAA quantification was performed based on standard curves prepared with pure IAA (BIOCHEM Chemopharma, Cosne-Cours-sur-Loire, France). All experiments were performed in triplicate.

Furthermore, we tested the effect of NaCl on IAA production by preparing LB media supplemented with L-Tryptophane and glucose (as previously described) with increasing NaCl concentrations (0; 100; 200; 300; 400; 500; 600; 700; 800; 900; 1000; 1100; 1200; 1300; 1400; and 1500 mM). After inoculation and incubation (in the same conditions as previously described), the amount of IAA produced was estimated spectrophotometrically at 530 nm (SHIMADZU UV-1800, Kyoto, Japan).

#### 2.2.3. Capacity for Solubilising Inorganic Phosphate

The halotolerant and IAA-producing strains were further tested for their ability to solubilize inorganic rock phosphate [Ca_3_(PO_4_)_2_], which was verified using Pikovskaya’s medium (PKV) containing in 1 L of distilled water: 10 g glucose; 0.5 g yeast extract; 5 g Ca_3_(PO_4_)_2_; 0.5 g (NH_4_)_2_SO_4_; 0.1 g MgSO_4_.7H_2_O; 0.2 g NaCl; 0.2 g KCl; 0.002 g MnSO_4_·H_2_O; 0.002 g FeSO_4_·7H_2_O; and 15 g of agar. After 3 days of incubation at 30 °C, the development of transparent halos around the colonies was considered as a positive result [26].

#### 2.2.4. Nitrogen Fixation

The halotolerant IAA-producing and phosphate solubilising strains were further tested for their ability to fix atmospheric nitrogen by growing them in selective N-free Jensen’s medium (1951) containing 1 L of distilled water: 20 g mannitol; 2.0 g CaCO_3_; 1.0 g K_2_HPO_4_; 0.5 g MgSO_4_.7H_2_O; 0.5 g NaCl; 0.1 g FeSO_4_; 0.005 g Na_2_MoO_4_; and 17 g agar. The medium pH was adjusted to 7.4 ± 0.02 [52]. 

#### 2.2.5. Enzymatic Activities

The halotolerant, IAA-producing, phosphate solubilising, and nitrogen-fixing strains were further tested for the following extracellular enzymatic activities: urease [53], esterase [54], lipase [55], cellulase [56], chitinase [57], amylase [58], and protease [59]. Enzymatic activities were tested on bacterial cultures growing in nutrient broth. These enzymes are involved in organic matter degradation and therefore increase soil fertility. Furthermore, the following enzyme activities may play a relevant role in plant defence: chitinase, cellulase [60], lipase [61], and protease [62,63].

Whenever a given enzymatic activity was detected, it was considered a PGP trait displayed by the halotolerant bacterial strain. A score, which is the sum of positive results for the enzymatic activities, was calculated for each halotolerant bacterial strain.

#### 2.2.6. Production of Plant Defence Compounds

The halotolerant, IAA-producing, phosphate solubilising, and nitrogen-fixing strains and with high scores of extracellular enzymatic activities were further tested for their ability to produce metabolites that are involved in plant defence, namely hydrogen cyanide (HCN) and ammonia (NH_3_). These metabolites have been shown to have antifungal activity, which is of great importance as plants under salt stress become more sensitive to phytopathogen attacks [38]. Ammonia can also be used by the plant as a nitrogen source. The production of HCN and NH_3_ was tested according to Lorck [64] and Ward et al. [65], respectively.

### 2.3. Molecular Identification of Selected Bacterial Isolates

The four halotolerant bacterial isolates that presented interesting plant growth-promoting traits were molecularly identified by sequencing the partial 16S rDNA gene. Genomic DNA extracts were obtained after incubating a single colony at 96 °C for 7 min. The polymerase chain reaction (PCR) was carried out in a total volume of 50 µL, containing 1x PCR buffer (Invitrogen, Waltham, MA, USA), 1.5 mM of MgCl2, 0.2 mM of each dNTP (Invitrogen), 1 µM of each primer (104F: 5′-GGACGGGTGAGTAACACGT-3′; 1392R: 5′-ACGGGCGGTGTGTRC-3′), 1U Taq DNA polymerase, and 2 µL of genomic DNA extract. Amplification conditions were as follows: initial denaturation of 3 min at 94 °C, 35 cycles of 1 min at 94 °C, 1 min at 55 °C and 1 min at 72 °C, and a final extension step of 3 min at 72 °C. Agarose gel electrophoresis (1.2%) was carried out in 1xTBE for 1 h at 90 V. Sequencing of the purified amplification products was performed in the reverse direction using primer 1392R.

Primary sequence analysis was carried out using Chromas Lite (Technelysium Pty Ltd., South Brisbane, Australia). The BLAST tool at the National Center for Biotechnology Information (NCBI) was used to identify the isolates to the genus level by comparing the obtained DNA sequences with publicly available sequence data. To achieve a more accurate phylogenetic characterisation, the obtained sequences were compared to a subset of sequences from the same genus using the Molecular Evolutionary Genetics Analysis MEGAX. After multiple sequence alignments using the Clustal algorithm, phylogenetic trees were generated using the neighbour-joining method. The reliability of the inferred phylogenetic trees was assessed using bootstrap analysis with 1000 replicates.

### 2.4. Testing the PGP Potential of the Selected Strains on Wheat

#### 2.4.1. Seed Germination

To test if those four bacterial strains could improve wheat germination under increasing salinity levels, our experimental design consisted of two factors: the halotolerant bacterial strains with PGP traits (strains S1, S2, S3, and S4 and the control without any bacterial inoculation) and increasing salinity levels (0, 150, and 250 mM of NaCl). The tested salinity levels correspond to: no salinity (0 mM of NaCl), high salinity (150 mM of NaCl), and extreme salinity (250 mM of NaCl) [49]. The design was fully factorial, resulting in 15 treatments with 5 replicates (Petri dishes) each (75 Petri dishes in total).

Bacterial suspensions of the identified strains (S1, S2, S3, and S4) were inoculated and incubated for 18 h with 0 mM of NaCl and then centrifuged at 3000 rpm for 10 min at 4 °C. The resulting pellets were washed three times with sterile sodium chloride solution (8.5 g L^−1^ of NaCl) and then suspended in the same solution (OD_600 nm_: 0.8) and used for germination tests. Non-damaged wheat (*Triticum aestivum* L.) seeds were surface sterilised with 70% ethanol for 1 min, followed by 5% HgCl_2_ for 3 min, and finally washed 10 times with sterilised distilled water. The surface-sterilised seeds were coated with bacterial strains by dipping them in the washed bacterial suspensions for 1 h with shaking at room temperature. The inoculated seeds were placed in Petri dishes (15 seeds per Petri dish of 9 cm diameter) containing growth medium with agar (0.8%) and increasing NaCl concentrations: 0, 150, and 250 mM. For each NaCl concentration, we included control seeds, which were not inoculated with any bacterial strain. Each combination (NaCl concentration and bacterial strain) was replicated 3 times (3 Petri dishes with 15 seeds each). Germination occurred in a growth chamber in the dark at 28 °C. Every two days, the number of germinated seeds was recorded until there were no more seeds germinating (12 days). The germination rate was calculated for each strain.

#### 2.4.2. Seedling Growth

To test if the bacterial strains could improve wheat seedlings’ survival, growth, and development under increasing salinity levels, our experimental design consisted of two factors: the halotolerant bacterial strains with PGP traits (strains S1, S2, S3, and S4 and the control without any bacterial inoculation) and increasing salinity levels (0, 150, and 250 mM of NaCl). The design was fully factorial, resulting in 15 treatments with 5 replicates (sterile tip boxes) each (75 sterile tip boxes in total).

Twelve days after germination, wheat seedlings were aseptically transferred from the Petri dishes from the corresponding combination of bacterial strain and salinity to a hydroponic system consisting of inverted sterile tip boxes (10 seedlings/box) containing ¼ Hoagland’s solution with the following composition in distilled water: 1.5 mM of KNO_3_; 1 mM of Ca(NO_3_)_2_·4H_2_O; 0.5 mM of NH_4_H_2_PO_4_; 0.25 mM of MgSO_4_·7H_2_O; 50 µM of KCl; 25 µM of H_3_BO_3_; 2 µM of MnSO_4_·H_2_O; 2 µM of ZnSO_4_·7H_2_O; 0.5 µM of CuSO_4_·5H_2_O; 0.5 µM of (NH_4_)_6_Mo_7_O_2_·4H_2_O; and 20 µM of FeNaEDTA. Hoagland’s nutrient solutions were prepared for the following NaCl concentrations: 0; 150, and 250 mM. All experimental setups were performed in triplicate. Wheat seedlings were grown under controlled conditions in a growth chamber (16 h/8 h light/dark; 25 °C/20 °C day/night; 350 μmol m^−2^ s^−1^).

After 14 days of the experiment, the plants were harvested, separated into roots and shoots, weighed (fresh weight) and dried to constant mass at 60 °C, and weighed again (biomass). Furthermore, other plant growth parameters were determined: root and shoot length (cm), root/shoot ratio, number of lateral roots, root surface area (mm^2^), root fresh weight (g), root dry weight (mg), shoot fresh weight (g), and shoot dry weight (mg).

### 2.5. Calculations and Statistics

Each combination of bacterial inoculation (including no inoculation) and NaCl concentration was normalised using the combination of no inoculation (i.e., control) and no salinity (i.e., 0 mM of NaCl) as references. This normalisation allowed us to calculate the inoculation effect (on germination and on biomass) using the control under 0 mM of NaCl (i.e., no inoculation, no salinity—0;0) as follows:$$\mathrm{Inoculation}\;\mathrm{effect}\;\left(\%\right)=\frac{\left(\mathrm{Parameter}\;0\ldots S4;\;0\ldots 250-\;\mathrm{Average}\;\mathrm{parameter}\;0;0\right)}{\mathrm{Average}\;\mathrm{parameter}\;0;0}\times 100$$

The effect of the agricultural field on soil physical and chemical parameters was tested separately using a one-way analysis of variance with site as the fixed factor. The effect of the halotolerant bacterial strains with PGP traits and increasing salinity levels on bacterial growth, wheat germination, and biomass were tested separately using a two-way analysis of variance with strain and salinity level as fixed factors. The inoculation effect on plant parameters (germination, biomass) was tested separately using a one-way analysis of variance with treatment (i.e., the combination of inoculation and NaCl concentration) as a fixed factor. Bonferroni post hoc multiple comparisons were used to test for differences (*p* < 0.05) in soil, while LSD (least significant difference) post hoc multiple comparisons were used to test for differences (*p* < 0.05) in plant parameters. SPSS (version 26⋅0, IBM, Inc., Chicago, IL, USA) was used for all the abovementioned analyses.

## 3. Results

### 3.1. Soil Physic-Chemical Characteristics

Soils from agricultural fields 1 and 2 only differed in salinity: soils from field 2 were considered as slight to moderate saline (EC > 4 dS m^−1^ at 25 °C and higher Na concentration), while soils from field 1 were not saline. Soils from both fields were alkaline with low organic matter (Table 1).

### 3.2. Screening for Halotolerant PGP Bacteria Candidates

By applying the screening sequence we used, we started with 37 morphologically distinct bacterial colonies isolated from the two agricultural sites. Selecting for the halotolerant strains (those which grew at NaCl concentrations > 100 mM) reduced the number to 22 strains. From those 22, only 15 produced IAA, and from those 15, only 12 solubilised inorganic phosphate and fixed atmospheric nitrogen. From those 12, only 4 presented high scores for extracellular enzyme activities. Therefore, from herein, only those four bacterial strains will be considered.

Furthermore, the four bacterial strains that kept being selected for their PGP traits (S1, S2, S3, and S4) displayed a remarkable halotolerance as they were able to grow under NaCl concentrations > 1000 mM (Figure 1). Two of these strains were isolated from the saline soil (agricultural field 2) and the other two from the non-saline soil (agricultural field 1) (Table 1). The strains isolated from the saline soil (S3 and S4) were the most halotolerant since they grew in up to 1400 mM of NaCl, while the strains isolated from the non-saline soil (S1 and S2) only grew in up to 1100 mM of NaCl (Figure 1).

Under no salinity (i.e., 0 mM of NaCl), the four strains produced similar amounts of indoleacetic acid (IAA): S1 produced 23 ± 1 µg mL^−1^; S2 produced 24 ± 2 µg mL^−1^; S3 produced 25 ± 3 µg mL^−1^; and S4 produced 28 ± 1 µg mL^−1^. Increasing the salinity up to 300 mM of NaCl stimulated IAA production: S2 doubled IAA production (52 µg mL^−1^); S3 almost tripled IAA production (70 µg mL^−1^); and S4 more than doubled IAA production (69 µg mL^−1^). S1, which showed the lowest stimulation at 300 mM of NaCl (37 µg mL^−1^), was able to produce IAA up to 700 mM NaCl (2 µg mL^−1^).

Physiologically, the four bacterial strains (S1, S2, S3, and S4) were similar in their ability to solubilise phosphate, fix atmospheric nitrogen, present high scores for extracellular enzyme activities, and produce plant defence compounds (Table 2). However, there were some small differences: strains S1 and S2, isolated from the non-saline soil, tested negative for urease activity, and strain S3, isolated from the saline soil, tested negative for esterase and could not produce HCN. Strain S4, isolated from the saline soil, was the only strain that tested positive for all 12 PGP traits.

### 3.3. Molecular Identification

A first BLAST analysis of each sequence allowed us to identify the genus of each isolate, and then a more thorough phylogenetic analysis, using the sequences of our isolates and the sequences of type strains of the most closely related species (retrieved from NCBI), allowed better allocation of each isolate within the genus in major clades. As the phylogenetic tree shows: (i) the strains isolated from the non-saline soil (agricultural field 1) belong to the *Pseudomonas fluorescens* lineage (S1 (OM281435)—Figure 2) and to the *Bacillus subtilis* clade (S2 (OM281438)—Figure 3) and (ii) the strains isolated from the saline soil (agricultural field 2) both belong to the *Bacillus megaterium* clade (S3 (OM281436) and S4 (OM281437)—Figure 3).

### 3.4. Testing the PGP Potential of the Selected Strains

#### 3.4.1. Wheat Seed Germination

Increasing salinity (i.e., NaCl concentrations) reduced wheat seed germination (Figure 4). Although there was a significant interaction between inoculation and salinity, inoculation with S4 (*B. megaterium* isolated from the saline soil) always promoted the best rates of germination for the three NaCl concentrations tested. Only the higher salinity level (i.e., 250 mM of NaCl) reduced the positive effect of inoculating the S4 strain on seed germination, which dropped from ~80% without salinity and with intermediate salinity (i.e., 0 and 150 mM of NaCl) to ~40% at 250 mM of NaCl. High salinity reduced the positive effect of the S4 strain on seed germination.

By contrast, S1 and S2 (*P. fluorescens* and *B. subtilis*, respectively), isolated from the non-saline soil, and S3 (*B. megaterium*), isolated from the saline soil, did not have an effect (positive or negative) on seed germination without salinity or with intermediate salinity (i.e., 0 and 150 mM of NaCl). However, high salinity (250 mM NaCl), did promote a positive effect of S1, S2, and S3 strains on seed germination.

#### 3.4.2. Wheat Seedling Growth

After 14 days of cultivation under the higher salinity level, all the control wheat seedlings (i.e., without any bacterial inoculation) were dead. By contrast, the wheat seedlings inoculated with the bacterial strains survived along the full range of salinity levels tested.

Salinity, but not bacterial inoculation, influenced seedling water content: the higher the salinity, the higher the water content (Appendix A). There were significant interactions between inoculation and salinity for several seedling growth parameters (e.g., biomass, root system, length—Table 3 and Figure 5 and Appendix A). Therefore, although increasing salinity had a negative effect on seedling growth parameters, in some cases, inoculation with some bacterial strains (especially S4) was able to overcome the negative effect of salinity. The positive effects of bacterial inoculation reflected more on roots than on shoots, as evidenced by higher root biomass, surface area and length, and the number of lateral roots in inoculated seedlings compared to control ones (Table 3 and Appendix A).

Increasing salinity reduced wheat seedling biomass (Figure 5). Although there was a significant interaction between inoculation and salinity, inoculation with S4 (*B. megaterium* isolated from the saline soil) always promoted higher biomass accumulation for the three NaCl concentrations tested. Since plant biomass remained high (~6 g seedling^−1^) without salinity and with intermediate salinity (i.e., 0 and 150 mM of NaCl, respectively), the positive effect of inoculating the S4 strain on plant biomass was only reduced under the higher salinity level (250 mM of NaCl). By contrast, S1 and S2 (*P. fluorescens* and *B. subtilis*), isolated from the non-saline soil, and S3 (*B. megaterium*), isolated from the saline soil, did not have an effect (positive or negative) on plant biomass without salinity or with intermediate salinity (i.e., 0 and 150 mM of NaCl). Since all control plants died under the higher salinity level (250 mM NaCl), inoculation with any bacterial strains had a positive effect on plant growth.

## 4. Discussion

By isolating halotolerant bacteria with PGP traits from both saline and non-saline soils where a salt-sensitive crop (i.e., wheat) was grown, our study clearly shows that: (i) non-halophyte plants and non-saline soils can also be reservoirs for halotolerant microbes and (ii) the PGP potential of halotolerant bacteria is not phylogenetically determined.

### 4.1. Are Bacterial Halotolerance and PGP Potential Shaped by Soil Salinity?

As the two agricultural soils only differed in electrical conductivity (EC) and sodium (Na) concentration (Table 1), it can be assumed that differences in salinity were the major driver of differences in the respective soil microbial communities (in terms of structure, abundance, and functioning). It is likely that the microbial community (and the bacterial one, in particular) in the saline soil had already been selected based on its halotolerance [43]. In agreement, as halotolerant bacteria may survive and maintain their metabolic functions under high salinity levels (Figure 1), the saline soil (i.e., agricultural site 2—Table 1) was shown to constitute a natural niche for halotolerant bacteria (i.e., 22 bacterial isolates were able to grow in salinity levels >100 mM of NaCl). However, 15 halotolerant bacterial strains were isolated from the non-saline soil (i.e., agricultural soil 1—Table 1). Soil heterogeneity and microstructure may help explain the presence of halotolerant bacteria in both saline and non-saline soils.

Soils are composed of micro-aggregates (<250 μm), which assemble into larger macroaggregates (0.25–2 mm) held together by organo-mineral complexes and encrusted organic matter, which create highly specific microenvironments so that contrasting microbial niches may co-exist in the soil [66]. Although the soils were classified as saline and non-saline (Table 1), as soils are very heterogenous [67], especially at the microbial/bacterial scale (i.e., the soil’s microstructure), it is likely that there were saline micro-sites in the non-saline soil, and vice versa. If this was the case, the halotolerant bacteria isolated from the non-saline soil (S1 belongs to the *P. fluorescens* clade (Figure 2) and S2 belongs to the *B. subtilis* clade (Figure 3)) were probably inhabiting saline micro-sites. Further supporting this hypothesis is the fact that these two bacterial strains displayed low growth rates for no- or low-salinity levels, clearly showing that increasing salinity up to 500 and 700 mM of NaCl stimulated their growth (Figure 1). Concurrently, these two bacterial strains only began to confer significant PGP benefits when the wheat seedlings were grown under the higher salinity level (250 mM) (Table 3 and Figure 4, Figure 5 and Appendix A). Therefore, the fact that a certain level of salinity may be required to promote bacterial growth and the full manifestation of its PGP potential may mean that the efficiency of a bacterial inoculant or biofertilizer may be modulated by the soil salinity level. 

However, strain S3 (belongs to the *B. megaterium* clade—Figure 3), isolated from the saline soil, also only began to confer significant PGP benefits when the wheat seedlings were grown under the higher salinity level (Table 3 and Figure 4, Figure 5 and Appendix A) but its growth was high under no- or low salinity (Figure 1). Therefore, in addition to the salinity effect on stimulating or inhibiting bacterial growth, other factors (e.g., the set of PGP traits) must have contributed to the differential PGP effect of S3 along the salinity gradient.

### 4.2. PGP Potential of the Selected Halotolerant Bacterial Strains

Salinity reduced plant growth and productivity of the salt-sensitive crop we tested (wheat, *Triticum aestivum*) [68]. However, as expected, inoculation with the halotolerant bacterial strains displaying PGP traits (Table 2) overcame the negative effects of salinity on wheat germination (Figure 4) and survival and growth (Table 3 and Figure 5 and Appendix A). Furthermore, only the inoculated wheat seedlings survived when grown under a higher salinity level. Not surprisingly, the four halotolerant bacterial strains displaying more PGP traits (Table 2) belong to genera that enclose most of the commercially available biofertilizers: *Pseudomonas* (S1, *P. fluorescens*—Figure 2) and *Bacillus* (S3 and S4, *B. megaterium*; and S2, *B. subtilis*—Figure 3).

Not all bacterial strains showed the same PGP potential along the salinity gradient we tested: strain S4 (*B. megaterium*), isolated from the saline soil, which displayed all the plant growth-promoting traits (Table 2), always conferred greater benefits to the plant host (Table 3 and Figure 4, Figure 5 and Appendix A) than the remaining strains, including the other bacterial strain isolated from the saline soil (S3,*B. megaterium*). The fact that these two bacterial strains both belong to the *Bacillus megaterium* clade (Figure 3) clearly shows that the PGP potential is not phylogenetically determined. Studies on the legacy of a farming system (conventional versus organic) on the physiology of phosphate solubilising bacteria and on the interactions among those bacteria also revealed that bacterial physiology and the output of the interactions (cooperation or antagonism) between the bacterial strains were not phylogenetically determined [26,69].

#### 4.2.1. Plant Growth-Promoting Traits Common to All Bacterial Strains

Since high salinity may decrease a plant’s endogenous phytohormones levels, seed priming with phytohormone-producing bacteria may constitute an important source of phytohormones [5,21,29]. The phytohormone most commonly produced by plant growth-promoting bacteria is indoleacetic acid (IAA), which is an auxin directly involved in plant growth promotion, commonly increasing germination and promoting the root system development (root elongation, lateral and adventitious root formation, and root hair formation), which improve plant water and nutrient uptake [70,71,72]. Therefore, the fact that all four bacterial strains produced IAA (Table 2) and that increasing salinity stimulated IAA production even further, must have played an important role in stimulating germination and promoting plant growth, especially during the first few days after seed sowing (Figure 4). Since IAA benefits manifest mainly in promoting root development, it is not surprising that inoculating the bacterial strains had more positive effects in the root system (increasing root biomass, surface and length, and the number of lateral roots; Appendix A) than in the shoots (shoot biomass and length) (Table 3).

Since N and/or P usually limit plant growth in terrestrial environments [37], the fact that all four bacterial strains presented traits capable of improving plant N and P nutrition will contribute to a positive effect on plant growth and development. Indeed, all four halotolerant bacterial strains were able to:

-Fix atmospheric N, which is an effective strategy for boosting plant development in salt-affected areas [5];-Produce ammonia, which can be used by the plant as a N source and can act as a biocontrol agent against pathogenic microorganisms (namely phytopathogenic fungi), which tend to proliferate under high salinity [38,41];-Solubilise phosphate that, despite being present in the soil, is mostly unavailable for plant uptake due to adsorption to soil particles and/or P immobilization [21,73]. The importance of halotolerant phosphate solubilising bacteria is even more relevant because high salinity causes phosphate precipitation, reducing the available phosphate even further [74].

Producing enzymes involved in nutrient cycling is another way that all four halotolerant bacterial strains displayed an ability to improve nutrient availability to the plant host (Table 2). For example, amylase (hydrolyses starch to diverse products and progressively originates smaller polymers of glucose units) [75] and cellulase (releases sugar units from the cellulose chain) [76] are involved in carbon cycling. In addition to being involved in nutrient cycling, chitinase, lipase, and protease also play important roles in plant defence: (i) chitinase together with cellulase are highly involved in biocontrol activity by degrading fungal cell walls [60]; lipase is part of the lipid-associated plant defence responses, which cleave or transform lipid substrates in various subcellular compartments [61]; and (iii) proteases act at the front line of defence and play pivotal roles in disease resistance [62,63] including in plant protection against herbivores [77].

#### 4.2.2. PGP Traits Not Displayed by Some Bacterial Strains

The three PGP traits that were not common to all halotolerant bacterial strains (Table 2) may help explain the differences in plant growth promotion we observed along the salinity gradient (Table 3 and Figure 4, Figure 5 and Appendix A). Despite being very closely related genetically (i.e., both strains belong to the *B. megaterium* clade—Figure 3) and displaying remarkable halotolerance (Figure 1), S3 and S4 differed in two of the tested plant-promoting traits, while S4 displayed all the tested PGP traits, and S3 was the only strain that lacked esterase activity and hydrogen cyanide (HCN) production (Table 2). Esterases are involved in the hydrolysis of short-chain acid triglycerides [54], creating free fatty acids, which are lactone precursors [78]. Given lactones’ importance in mediating microbe–microbe and plant–microbe communication [79,80], it is likely that bacterial strains without esterase activity (i.e., S3) will not produce lactone precursors, and therefore may not interact so closely with the plant host. Furthermore, as high salinity promotes some fungal soil-borne diseases in plants [38], the fact that S3 also did not produce HCN may have decreased its capacity as a biocontrol agent, especially against fungal phytopathogens through inhibition of mycelial growth [41].

Finally, urease activity was only lacking in the two halotolerant bacterial strains isolated from the non-saline soil (Table 2), which could have contributed to their positive effects on germination (Figure 4) and plant survival and growth (Table 3 and Figure 5 and Appendix A) under the higher salinity level. Bu et al. [81] observed that an *Arabidopsis* urease mutant displayed increased salt stress tolerance, and when the wild type (WT) was treated with a urease inhibitor, its salt stress tolerance was improved. Although urease activity generates ammonium, which is a N source for plants, it may exacerbate plant salt stress, which suggests that urease activity may not be a good trait for selecting halotolerant PGPB.

### 4.3. Perspectives

Since all four halotolerant bacterial strains with PGP traits had a positive effect on wheat germination (Figure 4), growth, and survival (Table 3 and Figure 5 and Appendix A), especially under the higher salinity level, they represent a potential tool to grow wheat in saline soils. In fact, the intermediate salinity level (150 mM of NaCl corresponds to almost 15 dS m^−1^) is classified as high salinity, and the high level (250 mM of NaCl corresponds to almost 25 dS m^−1^) is classified as extreme [49], which means that the four bacterial strains with PGP traits may enable farming in marginal lands seriously affected by salinity. Given that 50% of the total cultivated and irrigated agricultural land worldwide is affected by high salinity [82], there is a sizable market for bio-products with the ability to ameliorate crop yield under severe salinity. The generalised use of products with these characteristics would allow local populations to achieve food security and improve their income and well-being.

Furthermore, bacteria with multiple plant growth promoting traits, such as the four halotolerant bacterial strains used in this work (Table 2), are more likely to produce better and more consistent results than isolates with only one PGP trait [21]. As S4 (*Bacillus megaterium*) showed a remarkable halotolerance (grew up to 1400 mM of NaCl—Figure 1) withstanding high osmotic pressures, this may imply that it would be a suitable bacterium to survive the conditions of biofertilizer formulation and seed inoculation. Since drying procedures implemented in biofertilizer formulation are the main factors conditioning the viability of microbial cells, and later their effect when applied in the field [83], the capacity of halotolerant PGPBs to withstand high osmotic pressures should be further investigated as a fundamental trait when screening for biofertilizer candidates, even if they are not designed for saline conditions. Reinforcing this idea is the fact that S4 (*B. megaterium*) was the strain that promoted seed germination and plant growth the most along the salinity gradient, including the no salinity and the intermediate salinity levels (0 and 150 mM of NaCl, respectively).

Since the benefits to the plant host from many PGP microbes tend to be greater under stressful conditions [18,84,85], it is possible that inoculating these halotolerant bacterial strains in saline fields will provide more compelling evidence of their potential as candidates for biofertilizers than those obtained in this study. However, our results need to be validated with performance tests using soil under controlled (i.e., pot experiments in the greenhouse) and natural conditions (i.e., field experiments). Furthermore, more studies are required to explain the mechanisms involved in salt tolerance induction by these halotolerant bacterial strains in wheat seedlings, including assessing other plant growth-promoting traits such as (i) ACC deaminase, which has been shown to promote plant growth and development under adverse environmental conditions [86] and (ii) siderophore production, which have a high affinity with iron III from the rhizosphere and, consequently, retain a most of the iron available, inhibiting the proliferation of phytopathogenic fungi [26]. Finally, the nature of the compounds involved in salt tolerance induction by these halotolerant bacterial strains in wheat seedlings, which improved wheat growth under salinity, should be elucidated.

## 5. Conclusions

In addition to corroborating that saline soils constitute a natural niche for halotolerant microbes with PGP traits, our study shows that salt-sensitive plants and non-saline soils can also be good reservoirs for halotolerant PGPB. The presence of culturable halotolerant PGPB in both saline and non-saline soils likely reflects the importance of soil heterogeneity and microstructure in creating a wide range of soil micro-niches.

Not surprisingly, the four halotolerant bacterial strains displaying more PGP traits (related to phytohormone production, improved nitrogen and phosphorus availability, nutrient cycling, and plant defence) belong to genera that enclose most of the commercially available biofertilizers: *Pseudomonas* (one *P. fluorescens* strain isolated from the non-saline soil) and *Bacillus* (one *B. subtilis* strain isolated from the non-saline soil and two *B. megaterium* strains isolated from the saline soil). The effect of these four halotolerant PGPB strains on wheat germination, survival, and biomass along a salinity gradient showed that: (i) only the inoculated wheat seedlings survived when grown in the higher salinity level and (ii) one of the *B. megaterium* strains isolated from the saline soil was the halotolerant bacterial strain, which displayed all 12 PGP traits and promoted seed germination and plant growth the most. Since both *B. megaterium* strains isolated from the saline soil displayed a remarkable halotolerance but had different effects on plant performance, our data clearly show that the PGP potential is not phylogenetically determined.

Given the negative impact of drying procedures on microbial viability, the capacity of halotolerant PGPBs to withstand high osmotic pressures should be further investigated as a fundamental trait when screening for biofertilizer candidates, even if they are not designed for saline soils. Finally, our results need to be validated under controlled (i.e., pot experiments in the greenhouse) and natural conditions (i.e., field experiments), and more studies are required to explain the mechanisms and the nature of the compounds involved in salt tolerance induction.

## Figures and Tables

**Figure 1 microorganisms-11-01687-f001:**
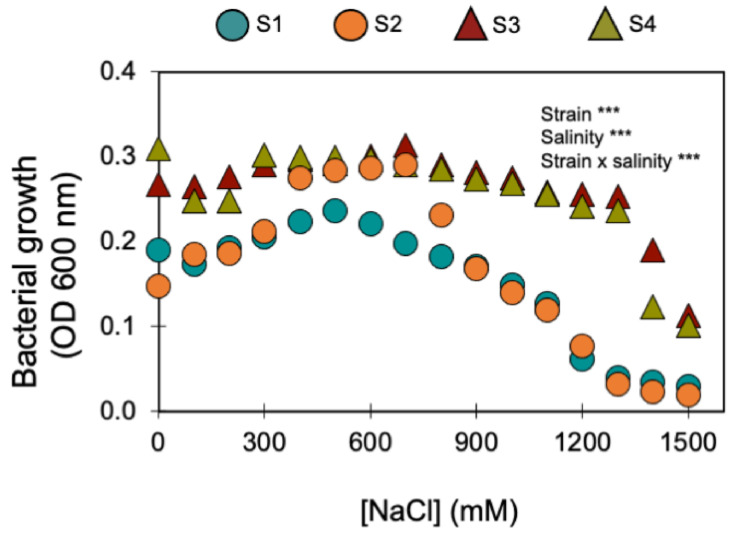
Effect of increasing salinity on the growth of the four halotolerant bacterial strains with PGP traits (circles show the bacterial strains isolated from the non-saline soil; triangles indicate the bacterial strains isolated from the saline soil). *** indicate significant effects (*p* < 0.01). Symbols are the mean ± 1SE (n = 3 replicates).

**Figure 2 microorganisms-11-01687-f002:**
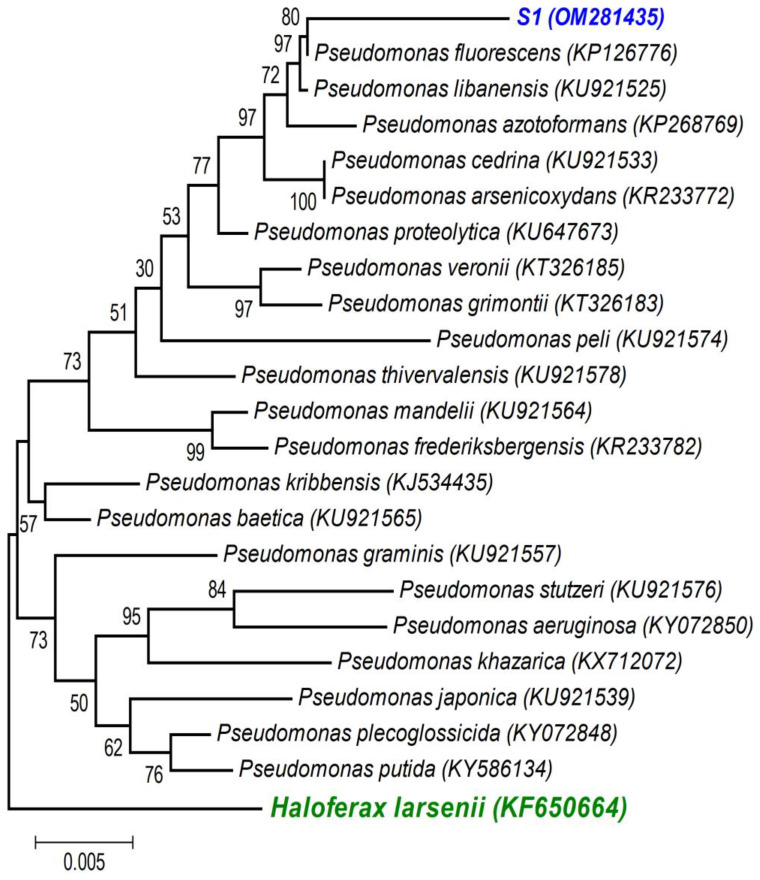
Neighbour-joining tree obtained using MEGA7, revealing the phylogenetic relationships for the studied *Pseudomonas* isolate (S1). (-) Bar represents sequence divergence of 1%.

**Figure 3 microorganisms-11-01687-f003:**
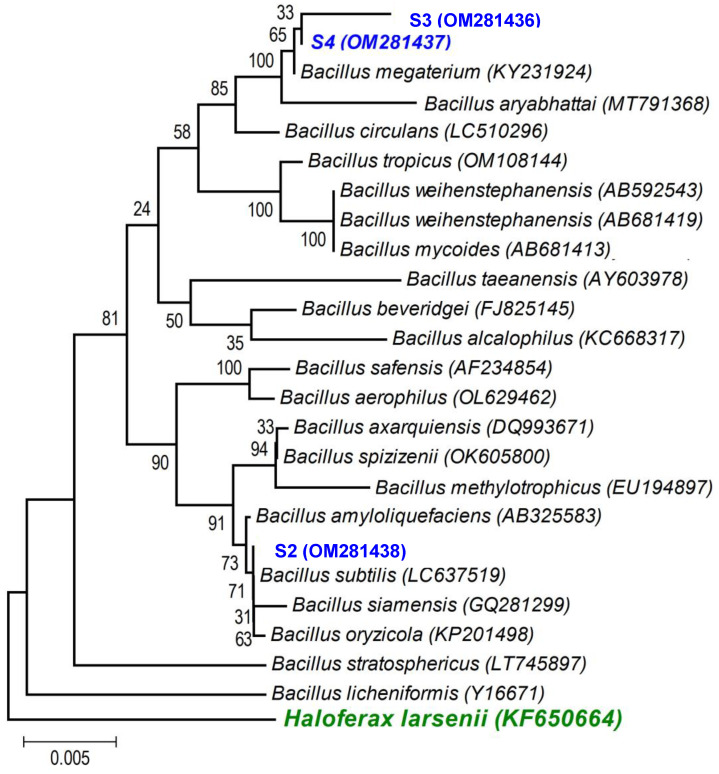
Neighbour-joining tree obtained using MEGA7, revealing the phylogenetic relationship for the studied *Bacillus* isolates (S2, S3, and S4). (-) Bar represents sequence divergence of 2%.

**Figure 4 microorganisms-11-01687-f004:**
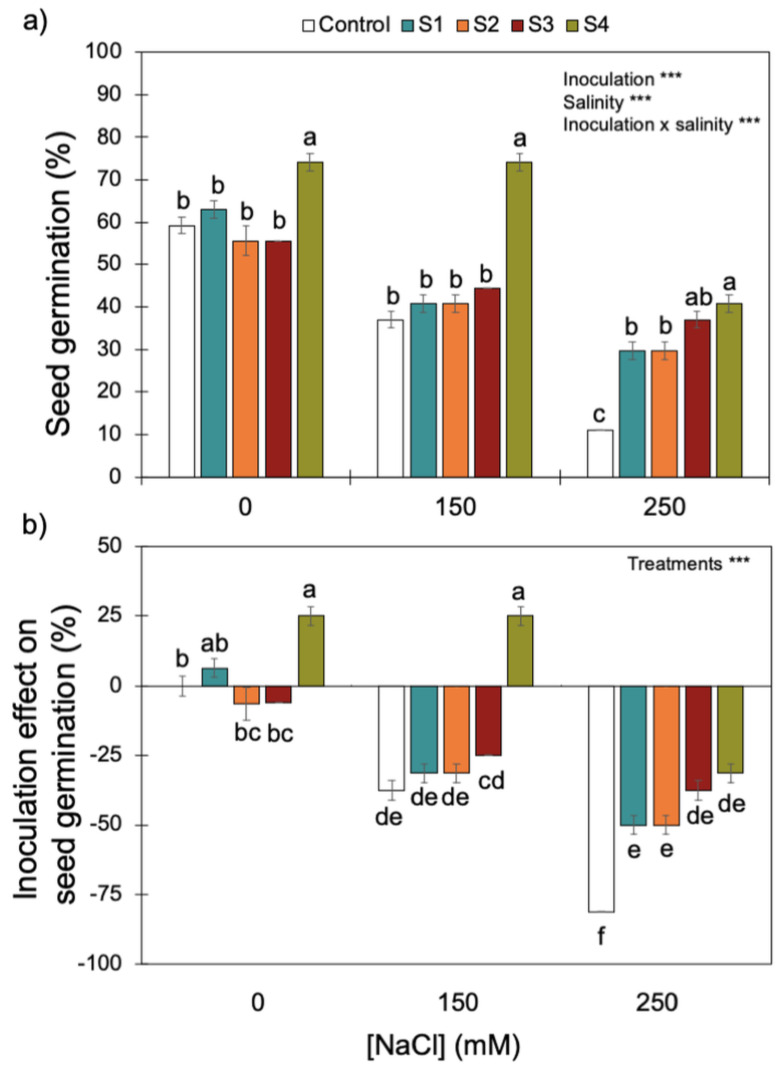
Effect of increasing salinity and bacterial inoculation on wheat germination (**a**) and the inoculation effect on germination (**b**)—please see Section 2). *** indicates significant effects (*p* < 0.01). Different letters indicate significant differences between (i) bacterial strains for each salinity level on graph (**a**) and (ii) treatments across all salinity levels on graph (**b**) (*p* < 0.05). Bars are the mean ± 1SE (n = 7 replicates).

**Figure 5 microorganisms-11-01687-f005:**
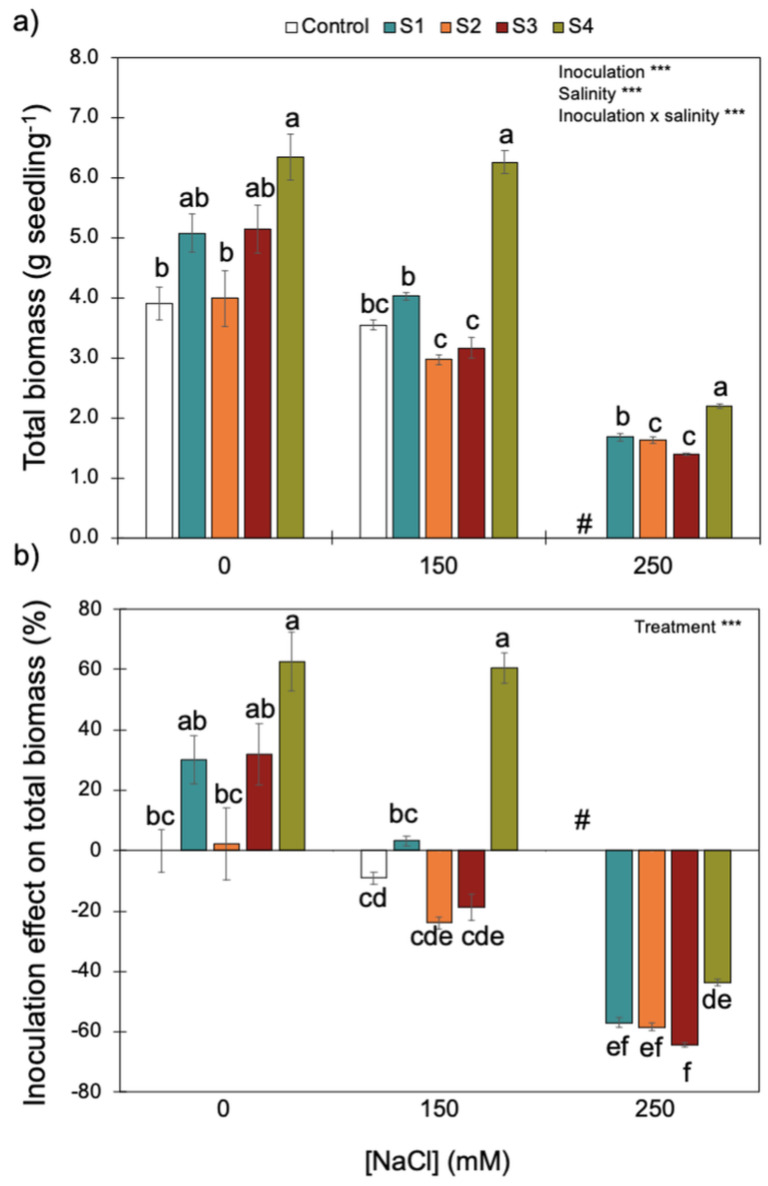
Effect of increasing salinity and bacterial inoculation on wheat seedling total biomass (**a**) and the inoculation effect on total biomass (**b**)—please see Section 2). *** indicates significant effects (*p* < 0.01). Different letters indicate significant differences between (i) bacterial strains for each salinity level on graph (**a**) and (ii) treatments across all salinity levels on graph (**b**) (*p* < 0.05). # indicates that for the control (no bacterial strain) under 250 mM of NaCl, all plants died. Bars are the mean ± 1SE (n = 7 replicates).

**Table 1 microorganisms-11-01687-t001:** Soil physical and chemical characteristics. Different letters indicate significant differences between agricultural fields (*p* < 0.05). Values are the mean ± SE (n = 3).

Soil Parameters		Agricultural Field 1	Agricultural Field 2
Salinity	EC (ds m^−1^)	2.35 ± 0.63 b	4.61 ± 0.02 a
	[Na] (%)	5.10 ± 1.90 b	10.15 ± 1.85 a
General properties	pH	7.95 ± 0.43	8.46 ± 0.04
	OM (%)	1.90 ± 0.40	1.05 ± 0.15
	Org C (%)	1.10 ± 0.20	0.65 ± 0.05
Micronutrients	Fe (ppm)	41.9 ± 14.7	42.1 ± 31.3
	Cu (ppm)	4.1 ± 3.0	4.0 ± 0.5
	Zn (ppm)	2.2 ± 0.2	1.9 ± 0.0
	Mn (ppm)	26.9 ± 2.2	24.0 ± 1.8

**Table 2 microorganisms-11-01687-t002:** Summary characterisation of the PGP traits displayed by the four ‘best candidates’ for halotolerant PGPB. ‘+’ and green shading means that the trait was observed in the bacterial strain, while ‘−’ and orange shading means that the trait was not observed in the bacterial strain.

		Non-Saline Soil		Saline Soil	
Benefit to the Host Plant	PGPB Traits	S1	S2	S3	S4
Phytohormones production	IAA production	+	+	+	+
Improved N availability	Nitrogen fixation	+	+	+	+
	Ammonia production	+	+	+	+
Improved P availability	Phosphate solubilisation	+	+	+	+
Enzymes involved in nutrient cycling	Amylase	+	+	+	+
	Cellulase	+	+	+	+
	Esterase	+	+	−	+
	Urease	−	−	+	+
Plant defence	Chitinase	+	+	+	+
	Lipase	+	+	+	+
	Protease	+	+	+	+
	HCN production	+	+	−	+

**Table 3 microorganisms-11-01687-t003:** Summary of the inoculation effects of each bacterial strain when compared to the control (without bacterial inoculation) for each salinity level on wheat seedling growth parameters (root biomass, surface area, number of lateral roots and length, and shoot biomass and length—please see Appendix A). ‘+’ and green shading means that inoculating the strain increased the parameter in relation to the control for the same salinity level, ‘−’ and orange shading means that inoculating the strain decreased the parameter in relation to the control for the same salinity level, and ‘ne’ and grey shading means that inoculating the strain did not change the parameter in relation to the control for the same salinity level.

			Root			Shoot		
Soil	Strain	[NaCl] (mM)	Biomass	Surface	Lateral Roots	Length	Biomass	Length
Non-saline	S1	0	+	ne	+	+	ne	ne
		150	+	ne	+	ne	ne	ne
		250	+	+	+	+	+	+
	S2	0	+	−	ne	ne	ne	ne
		150	+	ne	−	ne	−	−
		250	+	+	ne	+	+	+
Saline	S3	0	+	ne	+	ne	ne	ne
		150	+	−	−	ne	−	−
		250	+	+	+	+	+	+
	S4	0	+	+	+	+	ne	ne
		150	+	+	+	+	+	ne
		250	+	+	+	+	+	+

## Data Availability

Not applicable.

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
