# Peer review of "The Plant Growth-Promoting Potential of Halotolerant Bacteria Is Not Phylogenetically Determined: Evidence from Two Bacillus megaterium Strains Isolated from Saline Soils Used to Grow Wheat"

_microorganisms, 2023, doi:10.3390/microorganisms11071687_

Round 1

Reviewer 1 Report

It appears that this topic of research is of interest and would be suitable for publication in Microorganisms (ISSN 2076-2607). This manuscript examines the content of "The plant growth promoting potential of halotolerant bacteria is not phylogenetically determined: evidence from genetically similar yet functionally different strains" (microorganisms-2455560), which examined whether salt-sensitive crops can act as reservoirs for halotolerant bacteria that promote plant growth.

This study has some technical and structural issues that should be addressed during the revision process.

Following are the details of the comments.

1. The introduction needs to be revised. Some paragraphs require the reorganization of information. The content provided between lines 131 and 137 needs to be adjusted at the end of the introduction. Even though the authors tried to mention research objectives, these are not properly reflected in the contents of the results section. Furthermore, it is necessary to refine the research hypotheses.

2. Lines 332-349: Statistical analysis tools are not provided here. This information should be included.

3. Line 355: References are not used in the results section. It should be modified.

4. Line 660: Provide an independent conclusion section so that readers may easily be able to find the content of the conclusion.

5. The manuscript needs partial revision for language and grammar.

The manuscript needs partial revision for language and grammar.

Author Response

Dear Reviewer,

Many thanks for your comments and suggestions. Please find below our comments:

  1. The introduction needs to be revised. Some paragraphs require the reorganization of information. The content provided between lines 131 and 137 needs to be adjusted at the end of the introduction. Even though the authors tried to mention research objectives, these are not properly reflected in the contents of the results section. Furthermore, it is necessary to refine the research hypotheses.

Authors: we agree with the Reviewer and made several modifications along the manuscript and in the introduction in particular.

  1. Lines 332-349: Statistical analysis tools are not provided here. This information should be included.

Authors: we agree with the Reviewer and included the missing information.

  1. Line 355: References are not used in the results section. It should be modified.

Authors: we agree with the Reviewer and removed the reference from the results section.

  1. Line 660: Provide an independent conclusion section so that readers may easily be able to find the content of the conclusion.

Authors: we agree with the Reviewer and included a conclusion section.

  1. The manuscript needs partial revision for language and grammar.

Authors: we agree with the Reviewer and made the necessary changes.

Reviewer 2 Report

1.     Is the Title appropriate to describe whole story of the research? Can be Improved with more detailed and specific information related to the research. I think, your title right now shows more ambiguity to the readers and it will be better if you use effective words in the title.  

2.     Does the Abstract represent the research? Can be Improved by adding more background information why did you do your research.

3.     Do the authors summarize the main research question and key findings? Yes.

4.     In Introduction, Is the content succinctly described and contextualized with respect to previous and present theoretical background and empirical research (if applicable) on the topic? Can be improved by adding more information about the halotolerant bacteria.

5.     Do the authors identify other literature on the topic and explain how the study relates to this previously published research? Can be improved by making smooth connection among the information of the previous studies and your research in Introduction section. 

6.     Is the Objectives of the research address correctly? Can be improved by using more powerful words.

7.     Is the main question addressed by the research? No. Please add the main question of the research with relevant and interesting sentences. *Please double check the paragraph layout in the line 116 until 123.

8.     How original is the topic? The topic is original but I saw many studies with similar topic.

9.     Does the Methodology describe well? Yes

10.  Is the rationale for the proposed study clear and valid? Yes.

11.  Is the methodology technically sound? Will it effectively achieve its aims, and test the stated hypotheses? Yes.

12.  Is the methodology feasible and detailed enough to make the work replicable? Yes.

13.  Is the methodology and any analysis made correct and properly conducted? Yes.

14.  Are the experiments or interventions appropriate for addressing the research question? Yes.

15.  Is there enough data to draw a conclusion? Yes.

16.  Do the authors address any possible limitations of the research? Yes.

17.  Was data collected and interpreted accurately? Yes.

18.  Do the authors follow best practices for reporting? Yes.

19.  Does the study conform to ethical guidelines? Yes.

20.  Could another researcher reproduce the study with the same methods? In other words, have the authors provided enough information to validate the study? Yes.

21.  Is the statistical analysis adequate? Yes.

22.  Is the Result display in the correct way? Yes.

23.  Are the figures and tables clear and readable? Can be improved with higher resolution.

24.  Are the figure and table captions complete and accurate? Yes. It will be better if you also provide the microscopic pictures of all bacteria strains.

25.  Are the axes labeled correctly? Yes.

26.  Is the presentation appropriate for the type of data being presented? Yes.

27.  Do the figures and tables support the findings? Yes.

28.  Do the data provide enough evidence for the authors’ conclusions? Please add Conclusion section.

29.  Have the authors provided a sufficient amount of data and information for other researchers to recreate the analyses? Can be improved the detail of the data analyses.

30.  Does the Discussion describe all of the results? Yes.

31.  Are the arguments and discussion of findings coherent, balanced and compelling? Yes.

32.  Is the paper well written? Can be improved by checking error grammar, wrong diction, wrong punctuation, and also please make the story of the manuscript more smooth, so the reader can understand easily of your research.

33.  Is the text clear and easy to read? Can be improved by improving the correlation of your explanation among the paragraph. 

34.  Are the conclusions consistent with the evidence and arguments presented? No. Please add your conclusions with evidence and arguments. 

35.  Are the conclusions address the main question posed? Please add Conclusion section.   

36.  Are the conclusions supported by the data, and do they address the hypothesis? Please add Conclusion section.   

37.  Do the results support the conclusions? Yes.

38.  Does the Reference cite appropriately? Can be improved by adding the newest references.

39.  Are relevant data, citations, or references present? Yes.

Please double check the Grammar, Spelling, and Punctuation error.

Author Response

Comments and Suggestions for Authors

  1. Is the Title appropriate to describe whole story of the research? Can be Improved with more detailed and specific information related to the research. I think, your title right now shows more ambiguity to the readers and it will be better if you use effective words in the title.

Authors: we agree with the Reviewer and made some modifications that we consider will reduce ambiguity.

  1. Does the Abstract represent the research? Can be Improved by adding more background information why did you do your research.

Authors: although we agree with the Reviewer, the authors’ guidelines specify a limit of 200 words for the abstract. As it stands, our abstract already exceeds the word limit. Therefore, we only added one sentence.

  1. Do the authors summarize the main research question and key findings? Yes.

Authors: we appreciate the Reviewer´s opinion.

  1. In Introduction, Is the content succinctly described and contextualized with respect to previous and present theoretical background and empirical research (if applicable) on the topic? Can be improved by adding more information about the halotolerant bacteria.

Authors: we agree with the Reviewer and included a brief section on halotolerance and halotolerant bacteria.

  1. Do the authors identify other literature on the topic and explain how the study relates to this previously published research? Can be improved by making smooth connection among the information of the previous studies and your research in Introduction section.

Authors: we agree with the Reviewer and made some modifications.

  1. Is the Objectives of the research address correctly? Can be improved by using more powerful words.

Authors: we agree with the Reviewer and made some modifications.

  1. Is the main question addressed by the research? No. Please add the main question of the research with relevant and interesting sentences. *Please double check the paragraph layout in the line 116 until 123.

Authors: we agree with the Reviewer and made some modifications.

  1. How original is the topic? The topic is original but I saw many studies with similar topic.

Authors: we agree with the Reviewer and the reason why there are many studies on this topic is because salinity is a big challenge for agriculture and food security. Indeed, the growing pace of increasing salinity is pushing researchers to find solutions.

  1. Does the Methodology describe well? Yes

Authors: we appreciate the Reviewer´s opinion.

  1. Is the rationale for the proposed study clear and valid? Yes.

Authors: we appreciate the Reviewer´s opinion.

  1. Is the methodology technically sound? Will it effectively achieve its aims, and test the stated hypotheses? Yes.

Authors: we appreciate the Reviewer´s opinion.

  1. Is the methodology feasible and detailed enough to make the work replicable? Yes.

Authors: we appreciate the Reviewer´s opinion.

  1. Is the methodology and any analysis made correct and properly conducted? Yes.

Authors: we appreciate the Reviewer´s opinion.

  1. Are the experiments or interventions appropriate for addressing the research question? Yes.

Authors: we appreciate the Reviewer´s opinion.

  1. Is there enough data to draw a conclusion? Yes.

Authors: we appreciate the Reviewer´s opinion.

  1. Do the authors address any possible limitations of the research? Yes.

Authors: we appreciate the Reviewer´s opinion.

  1. Was data collected and interpreted accurately? Yes.

Authors: we appreciate the Reviewer´s opinion.

  1. Do the authors follow best practices for reporting? Yes.

Authors: we appreciate the Reviewer´s opinion.

  1. Does the study conform to ethical guidelines? Yes.

Authors: we appreciate the Reviewer´s opinion.

  1. Could another researcher reproduce the study with the same methods? In other words, have the authors provided enough information to validate the study? Yes.

Authors: we appreciate the Reviewer´s opinion.

  1. Is the statistical analysis adequate? Yes.

Authors: we appreciate the Reviewer´s opinion.

  1. Is the Result display in the correct way? Yes.

Authors: we appreciate the Reviewer´s opinion.

  1. Are the figures and tables clear and readable? Can be improved with higher resolution.

Authors: we agree with the Reviewer and improved the figures’ resolution.

  1. Are the figure and table captions complete and accurate? Yes. It will be better if you also provide the microscopic pictures of all bacteria strains.

Authors: we agree with the Reviewer that microscope photographs of the bacterial strains would be interesting. However, we do have access to a microscope capable of taking good photographs (e.g., scanning electron microscope).

  1. Are the axes labeled correctly? Yes.

Authors: we appreciate the Reviewer´s opinion.

  1. Is the presentation appropriate for the type of data being presented? Yes.

Authors: we appreciate the Reviewer´s opinion.

  1. Do the figures and tables support the findings? Yes.

Authors: we appreciate the Reviewer´s opinion.

  1. Do the data provide enough evidence for the authors’ conclusions? Please add Conclusion section.

Authors: we agree with the Reviewer and included a conclusion section.

  1. Have the authors provided a sufficient amount of data and information for other researchers to recreate the analyses? Can be improved the detail of the data analyses.

Authors: we agree with the Reviewer and made some modifications.

  1. Does the Discussion describe all of the results? Yes.

Authors: we appreciate the Reviewer´s opinion.

  1. Are the arguments and discussion of findings coherent, balanced and compelling? Yes.

Authors: we appreciate the Reviewer´s opinion.

  1. Is the paper well written? Can be improved by checking error grammar, wrong diction, wrong punctuation, and also please make the story of the manuscript more smooth, so the reader can understand easily of your research.

Authors: we agree with the Reviewer and made some modifications along the manuscript.

  1. Is the text clear and easy to read? Can be improved by improving the correlation of your explanation among the paragraph.

Authors: unfortunately, we cannot understand the Reviewer’s suggestion. Nevertheless, we hope that the modifications made along the manuscript improved its readability and clarity.

  1. Are the conclusions consistent with the evidence and arguments presented? No. Please add your conclusions with evidence and arguments.

Authors: we agree with the Reviewer and made some modifications that we consider will reduce ambiguity.

  1. Are the conclusions address the main question posed? Please add Conclusion section.  

Authors: we agree with the Reviewer and included a conclusion section.

  1. Are the conclusions supported by the data, and do they address the hypothesis? Please add Conclusion section.  

Authors: we agree with the Reviewer and included a conclusion section.

  1. Do the results support the conclusions? Yes.

Authors: we appreciate the Reviewer´s opinion.

  1. Does the Reference cite appropriately? Can be improved by adding the newest references.

Authors: we agree with the Reviewer but we already have 86 references which we consider are well balanced between recent findings and the first studies, especially those reporting methods which are still used. Therefore, we did not change this manuscript’s references.

  1. Are relevant data, citations, or references present? Yes.

Authors: we appreciate the Reviewer´s opinion.

Round 2

Reviewer 1 Report

The revised manuscript has improved significantly.